# Precision Technologies to Address Dairy Cattle Welfare: Focus on Lameness, Mastitis and Body Condition

**DOI:** 10.3390/ani11082253

**Published:** 2021-07-30

**Authors:** Severiano R. Silva, José P. Araujo, Cristina Guedes, Flávio Silva, Mariana Almeida, Joaquim L. Cerqueira

**Affiliations:** 1Veterinary and Animal Research Centre (CECAV), Associate Laboratory of Animal and Veterinary Sciences (AL4AnimalS), University of Trás-os-Montes e Alto Douro, Quinta de Prados, 5000-801 Vila Real, Portugal; ssilva@utad.pt (S.R.S.); cguedes@utad.pt (C.G.); fsilva@uevora.pt (F.S.); mdantas@utad.pt (M.A.); 2Escola Superior Agrária do Instituto Politécnico de Viana do Castelo, Rua D. Mendo Afonso, 147, Refóios do Lima, 4990-706 Ponte de Lima, Portugal; pedropi@esa.ipvc.pt; 3Mountain Research Centre (CIMO), Instituto Politécnico de Viana do Castelo, Rua D. Mendo Afonso, 147, Refóios do Lima, 4990-706 Ponte de Lima, Portugal

**Keywords:** dairy cows, welfare, precision livestock farming, lameness, mastitis, body condition score, behavior, infrared thermography

## Abstract

**Simple Summary:**

The welfare of farm animals is a growing concern in the EU and across the world. In milk production, there is a strong need to assess the welfare of dairy cows. One of the most sound assessment initiatives has been practiced using protocols developed by the Welfare Quality project. These protocols mainly support the assessment of cow welfare with animal-based indicators. However, evaluating these indicators is time-consuming and expensive, so using precision livestock farming (PLF) solutions is a way forward and is becoming a reality in the dairy industry. This review presents advances in PLF solutions, particularly in the last five years, and for assessing the animal-based indicators of lameness, mastitis, and body condition in dairy cattle farming.

**Abstract:**

Specific animal-based indicators that can be used to predict animal welfare have been the core of protocols for assessing the welfare of farm animals, such as those produced by the Welfare Quality project. At the same time, the contribution of technological tools for the accurate and real-time assessment of farm animal welfare is also evident. The solutions based on technological tools fit into the precision livestock farming (PLF) concept, which has improved productivity, economic sustainability, and animal welfare in dairy farms. PLF has been adopted recently; nevertheless, the need for technological support on farms is getting more and more attention and has translated into significant scientific contributions in various fields of the dairy industry, but with an emphasis on the health and welfare of the cows. This review aims to present the recent advances of PLF in dairy cow welfare, particularly in the assessment of lameness, mastitis, and body condition, which are among the most relevant animal-based indications for the welfare of cows. Finally, a discussion is presented on the possibility of integrating the information obtained by PLF into a welfare assessment framework.

## 1. Introduction

Animal welfare has long been considered a high priority within the European Union (EU), with several legislative initiatives from the late 1980s to the present day [1]. In parallel, the EU has invested significantly in research into farm animals’ welfare as part of a policy-oriented approach to identifying ways to improve animals’ lives [2,3]. Animal evaluation is an essential part of improving the standard of animal welfare. In this sense, efforts have been made to research science-based welfare indicators as assessment tools [4]. For example, the Welfare Quality^®^ project contributed with protocols to assess animal welfare in cattle, pigs, and poultry [5,6]. A few years later, the AWIN^®^ project produced indicators for species not considered in Welfare Quality^®^, namely horses, donkeys, turkeys, sheep, and goats [7]. However, there are many practical challenges in applying these protocols, which prevent them from having the maximum impact on the quality of life of farm species [8,9,10]. Nevertheless, the developments achieved in the last two decades in precision livestock farming (PLF), with close collaboration between researchers associated with engineering and the livestock sector, have driven a significant evolution in animal welfare assessment. PLF has developed rapidly in recent years, and animal welfare can be objectively assessed in real-time using a wide variety of indicators [11]. This assessment of welfare indicators is already possible, and it is expected to undergo extraordinary progress in the near future for livestock production. This will require the use of the latest developments in information, communication, and sensor technology [12]. Monitoring the welfare of cows, their productivity, and management practices is achievable through data from image, sound, and movement sensors that are combined with algorithms [13,14]. At the moment, there is robust knowledge that points to the possibility of monitoring and evaluating welfare automatically and with outputs that can be integrated into welfare protocols [11,15,16]. Additionally, an appropriate data visualization is necessary, so that farmers have a good acceptance of and efficiently use the technologies in PLF solutions [17].

In this review, an analysis will be made of the recent work of PLF in evaluating lameness, mastitis, and body condition, which are considered welfare indicators for dairy cows. It was also the objective of this review to point out future perspectives for PLF solutions, to automatically include animal-based indicators in a dairy farm welfare framework, allowing for the creation of better welfare for the animals and value for the farmer.

## 2. Welfare of Dairy Cows and Precision Livestock Farming

Currently, there are three welfare evaluation systems for dairy cattle, farmers assuring responsible management in USA [18], the code in New Zealand [19], and welfare quality in Europe [20]. The latter system has been seriously disputed in several reports [21,22,23], which presented several suggestions for reducing the number of evaluated parameters to overcome the time-consuming observations, which is a constraint that limits its routine application in dairy farms. In addition to shortening the assessing procedures, the method of calculating the scores was also changed and made more flexible, so that measures may be substituted or added as considered appropriate [22]. Another welfare evaluation system in development, according to Krueger et al. [24], is the integrated diagnostic welfare system (IDWS). This system might address some of the shortcomings of the other three systems, because it uses technology to help farms in the evaluation of animal welfare and to identify any causes of poor welfare. However, a considerable amount of data and records are needed to record animal behavior, health, and welfare conditions; and the use of sensors and technology can help in this matter [25]. According to Knight [26], research on dairy cow sensors has been very dynamic for assessing lameness, mastitis, and body condition, which will be the focus of this work. However, the application of sensors is extended to many other targets, such as aspects of reproduction (e.g., estrous cycle and parturition), nutrition, health, and general management. In this way, the main monitoring systems in dairy farms provide comprehensive information in different areas and demonstrate their suitability and feasibility for application on the dairy farm [25].

### 2.1. Lameness

Lameness is ranked as the third most important cause of economic losses on dairy farms, after mastitis and reproduction disorders. Lame cows are more frequently affected by mastitis, metabolic disorders, and reduced fertility [27]. In dairy cows, lameness can vary significantly in severity and can arise weeks, or even months, after a metabolic disorder, making the detection of causality complex [28]. Lameness is usually detected when the disease is already at an advanced stage and requires immediate and often expensive treatment. An animal in these circumstances can take several weeks to recover, representing a high cost for dairy farmers in terms of time, financial expenses for calls to the veterinarian, medication and treatment [29]. Time limitations of the dairy producers is a factor that contributes to the under-detection of lameness problems. Therefore, using flexible and affordable sensor-based systems is a need for recording the cows' behavior and thus identify the onset of lameness [30]. Lameness management consists of both prevention and treatment. Prevention management is linked with factors that are associated with lameness, such as improving walking surfaces, nutrition, and genetics. For a lame cow to be treated, it must first be identified as lame by the farmer. This generally occurs in three ways. The first is using a locomotion scoring system to systematically assess the herd [31]. The second is routine hoof trimming. Here, legs are lifted, inspected, and, if required, treated [32]. The third and most common is ad hoc observation during other activities, such as herding. Unfortunately, ad hoc detection is ineffective at detecting mild and even moderate lameness.

Automated lameness detection could provide useful cow and herd information addressing an information gap, particularly for mild and moderately lame cows. Earlier detection and automatic drafting could reduce the time from lameness onset to treatment, preventing cases from becoming severe, speeding up recovery, increasing production, and improving welfare [33]. In addition, lame cows tend to spend less time eating, with shorter bouts, and eat less during the day [34,35]. Automated lameness identification costs may be prohibitive, depending on the system. Nevertheless, to increase the cost-effectiveness of automatic systems, it is necessary to proceed with the downscaling of the current systems to increase the sensor detection performance and further enhance the system for other physiological states such as estrus, disease, calving, or body condition score (BCS) [36]. The single accelerometer per cow approach is particularly promising from a cost perspective, but several hurdles remain before such technology can be widely adopted on the farm. The foremost of these is developing reliable indicators of lameness using only one low or medium resolution pedometer. According to Schlageter-Tello et al. [37], most automatic locomotion scoring systems attempt to mimic human observers by measuring and analyzing cows’ locomotion and behavior parameters through sensors and mathematical algorithms. The technologies employed can be grouped into kinematic (pressure plate/load cell solutions, image processing techniques, and activity-based techniques); kinetic (ground reaction force systems, force-scale weighing platforms, and kinetic variations of accelerometers); and indirect methods, which mainly include behavior technologies and individual cow milk production measuring technologies.

#### 2.1.1. Pressure Plate/Load Cell

In pressure plate/load cell solutions, the aim is to examine how the weight is distributed across the legs of the animal as it walks through pressure-sensitive equipment. Stance time asymmetry, as measured by a Gaitwise pressure sensor [38], and three-dimensional force plate measurements of hind legs [39] have been identified as approaches for identifying cow lameness. Van Nuffel et al. [40] reported that stride length (meters) and duration (seconds) were indicative of lameness using the Gaitwise pressure mat system. Using the Gaitwise system, stance time (weight-bearing) for the non-lame leg was also found to be longer in lame cows [31]. Lame cows are cautious about placement of the affected foot, as this action is painful [41]. These authors reported that the duration of foot placement and foot lifting was relatively longer for lame cows. The disadvantage of the Gaitwise system compared to other image-based systems is the larger space needed for installation and the system cost. To reduce the cost, 14 configurations were studied to simulate the effects of decreasing mat length and sensor resolution [41]. The results showed that the length can be reduced by about 33% (4.88 to 3.28 m), while the downscaling of the sensor resolution by up to four times the original resolution was possible without decreasing the lameness detection performance for successfully monitoring one complete gait cycle [41]. Table 1 reports a summary of research work for assessing the lameness of dairy cows by kinematic and kinetic approaches.

#### 2.1.2. Image Processing Techniques

Image processing techniques analyze the posture of the animal as it walks through an alley or to a milking parlor. Solutions with 2D or 3D video cameras have the potential to be applied in lameness monitoring systems. Considering the character individual of normal and lame walking of the cows, however, challenges arise with the development of algorithms that must work broadly for all cows. Real-time lameness detection systems must consider normal and healthy behavior to detect abnormalities immediately to overcome this challenge. Typically in the 2D and 3D image system, the back posture is examined to measure the degree of lameness, and values are automatically extracted from a top view of the cows [64]. However, as mentioned previously, the back posture shows individual cow variation, indicating lameness for one cow but normal gait for another. Thus, cow posture values must be analyzed individually and compared with what is considered normal for each cow separately. The analysis of historical and real-time data from a given animal allows tuning a model to a healthy reference behavior in the case of lameness monitoring [65]. In addition, to overcome the inaccurate detection of lameness due to the individual characteristics of cows, Kang et al. [66] successfully studied (accuracy of 96%) a lameness detection method based on the supporting phase using computer vision. Van Hertem et al. [64] achieved a high specificity of 94.1%, which means that their algorithm generated minimal false alarms, a very desirable trait in lameness detection systems. Table 2 summarizes the research works assessing the lameness of dairy cows using 2D and 3D sensors.

#### 2.1.3. Activity-Based Techniques

Activity-based techniques typically use accelerometers (2D and 3D) to record the movement patterns of the animal. The data is then used to build the daily activities of the cow, e.g., walking and lying down. In a recent comprehensive work on detecting lameness in cattle, O'Leary et al. [75] support results from another report [76] that show the length of lying time is not a reliable indicator because it only explained a small proportion of the variation of lameness in dairy cows as lying time is influenced by many other factors. For these reasons, further research to support automatic lameness detection needs to focus on aspects other than lying time measures to succeed [75]. In this sense, other authors [77] developed a model for automatic lameness detection using data from an accelerometer-based approach applied to multiparous Holstein lame (n=41) and non-lame (n=12) cows. This work showed that lame cows show shorter strides and a slower walking speed than non-lame cows and that the best model to detect cows being lame considers the number of standing bouts and walking speed with a sensitivity of 90.2% and specificity of 91.7%. Also, measuring acceleration at the metatarsal level with accelerometers in each of the hind limbs proved to be a promising tool to describe the different variables of the gait cycle accurately [44]. In recent years, the development of accelerometer-based automated lameness detection systems has continuously evolved [75]. The first system was marketed in October 2018 by IceRobotics (Edinburgh, UK) [78]. In this system, each cow is equipped with a single low-resolution accelerometer. The system presents users with simple information similar to the traffic light system with the colors green, yellow and red if the cows are identified as likely to be non-lame, maybe lame, or those likely to be lame, respectively [78]. This approach can be very suitable for straightforward communication to farmers [75]. Another lameness detection system that shows a good trade-off between sensitivity and specificity is the combination of different sensor data, including milk yield, neck activity, and rumination time, which can perform with a sensitivity of 89%, a specificity of 85%, and an accuracy of 86% [64].

#### 2.1.4. Behavior of the Cows

Behavior assessment has played a huge role in evaluating animal welfare [79,80], including for dairy cows [81,82,83]. Since behavior assessment can be a long-term task, the use of technology is crucial [16]. Evaluating change in an animal's behavior is one of the most used criteria to assess its health and welfare. A good example is given by the pain linked with diseases of the claws or limbs of dairy cows, which produce changes in movement pattern and a decrease in daily activity [77]. Using diverse sensor types in different body locations (e.g., neck or leg-mounted) would be required to correctly classify lying, standing and feeding, which are key behaviors in dairy cows [30,84]. For example, Barker et al. [30], who used automated behavioral data collection through a combined position and activity sensor, observed a shorter feeding duration for lame cows than non-lame cows. This result shows that behavior analysis can be a tool for monitoring the health and welfare of cows [30]. The accelerometers can provide an indirect measure of the flinch, step, and kick (FSK) response. This information, combined with remote sensing of FSK, and integrated into existing systems where other production and behavioral information is available (e.g., the number of visits, feed intake, milk yield), could provide a non-invasive, real-time assessment of animal health and welfare. Combined with other data using infrared thermography (IRT), an automated system may be able to identify animals with the early onset of pathological or metabolic diseases and distress or discomfort, allowing an early intervention by the farmer and improving animal health, production, and welfare [84,85].

### 2.2. Mastitis

One of the most relevant diseases in dairy cows is mastitis, a cause of suffering in infected animals, with worldwide recognized harmful effects on the welfare and profitability of dairy farms [86,87]. Thus, producers have been concerned with implementing effective methods to control mastitis in their herds since the first mechanized milking systems appeared. The development and implementation of control programs that integrate pre and post-milking teat immersion, correct milking procedures and restricted use of antibiotics in drying only in infected cows have resulted in a significant decrease in contagious pathogens. However, as mastitis pathogens emerged, researchers sought to restrict the use of antimicrobials while preserving animal welfare and respecting universal guidelines for unnecessary use. Thus, despite remarkable advances in mastitis control during the last decade, mastitis will remain an important focus of future research [88]. 

Reliable detection of mastitis through automated methods represents an excellent opportunity to carry out early treatment programs and avoid overuse of antibiotics, preserving the health and welfare of cows, avoiding discomfort and pain, improving the recovery rate and the economic sustainability of farms [89,90]. Effective diagnostic methods can lead to faster and more efficient mastitis control and promote responsible use of antimicrobials [91]. It is also essential to reliably score the severity of clinical mastitis to predict treatment outcomes [92] and adapt treatment protocols accordingly. 

#### 2.2.1. Somatic Cell Count (SCC)

Health management is essential for maintaining efficient and sustainable dairy production. Somatic cell count (SCC) is the most used indicator to assess udder health status in dairy cows, being used at a quarter, cow and bulk tank levels. In automatic milking systems (AMS), fully automated online analysis equipment is available to analyze SCC at the farm at each milking [93]. Moreover, from the results of the online SCC, a number of additional cows and quarter level factors important for udder health are recorded in these systems [94]. The SCC can, to some extent, be used for the surveillance of intramammary infection, and the industry has advanced toward developing new sensors that are designed explicitly for udder health surveillance. One of these is the DeLaval Online Cell Counter (DeLaval, Tumba, Sweden), which allows repeated measurements of cell counts at the cow level. These may be implemented in automated detection systems to manage udder health in AMS [95]. This represents a considerable increase in the amount of data, for example, for udder health management, which can also serve as phenotypes for reproductive programs. In addition to the frequent measures of SCC, a number of additional cow level and quarterly factors considered of importance for udder health are recorded in the AMS in each milking [96].

#### 2.2.2. Electrical Conductivity and Lactate Dehydrogenase

Electrical conductivity (EC) and enzymatic concentrations of lactate dehydrogenase (LDH) have been used as indicators to detect mastitis [97,98]. Recent works have shown the potential of using sensors for automatic measurement of EC and LDH; however, the results showed that there is still a need for further research in this field [96]. In recent years, there has been an increasing choice of AMS worldwide. This type of equipment allows producers to increase milking frequency, milk production per cow and reduce labor costs [99]. The AMS is equipped with in-line sensors that measure EC to detect mastitis. These sensors make a continuous assessment of the concentration of milk ions during the milk collection process. However, the results are variable, with the first milk collected before milk ejection being more sensitive to detecting mastitis than the first harvested milk, which is explained by udder preparation and teat cleaning in AMS systems [100]. For this reason, it is pointed out that in the future, to improve the ability of AMS to detect mastitis, sensors should monitor the milk before teat cleaning [100].

#### 2.2.3. Infrared Thermography

Infrared thermography (IRT) is a non-invasive technology that allows accurate temperature measurement from a distance with a wide application in animal science [101,102]. In dairy production, IRT has been successfully used for early mastitis detection. Despite the proven ability to detect mastitis, there are limitations in the manual analysis of animals because this is time-consuming and requires a skilled examiner [103]. Zaninelli et al. [104] used software that located the pixel with the highest temperature in udder thermograms to distinguish between cows with normal and elevated SCC. Automatic evaluation of thermograms of bovine udders that received an intramammary challenge with E. coli showed promising results for detecting clinical mastitis, and these results were valid compared with the current gold standard of manual evaluation. We presume that the higher temperatures observed using manual analysis occurred because warmer regions were included, such as the udder–thigh cleft, whereas automatic segmentation omits these regions [103]. This method may also detect changes in the inner core temperature, such as fever. However, infrared thermography is intended for use as an automatic health surveillance tool and should not replace the examination of individual animals [105].

### 2.3. Body Condition Scoring

Body condition is a significant welfare and herd management indicator. Body condition is in high correlation with the health and metabolic status of the dairy cow and also with milk composition during lactation [106]. Body condition assessment is an indirect appraisal of the level of body reserves, and deviations reveal aggregate variation in energy balance [107,108]. The routine evaluation of body condition is based on visual observation and palpation of specific body areas to determine a score that assesses the adipose tissue and muscle mass deposits [109]. This assessment approach, generally known as the body condition score (BCS), has justified attention as a relevant tool for managing dairy herds [110]. 

BCS assessment can be performed by visual assessment or by a combination of visual indicators with palpation of bone structures and the degree of subcutaneous fat. The key areas for BCS assessment are the backbone, pins, tail head, long ribs, short ribs, hips, and rump [106]. Over the years, different scoring scales have been developed around the world. For example, a five-point scale system was commonly used in the USA, proposed by Windman et al. [111]. For its part, Ferguson et al. [112] proposed a scale of 0 to 5, subdivided into 0.25 centesimal, which assesses the body condition, particularly the adipose tissue of the cow's lumbar and pelvic areas. Despite the general agreement of dairy producers, nutritionists, and herd managers about the benefits of BCS evaluation, some factors discourage the use of traditional BCS evaluation techniques [113]: subjectivity in judgment can lead to different scores for the same cow under consideration, and the complex and time-consuming on-farm training of technicians [107]. Moreover, to have valuable information, cow measurements must be collected every 30 d throughout the lactation cycle [114], thus increasing the cost and complexity of collecting BCS data. To overcome these limitations, several solutions have been developed within the scope of the PLF that have shown very encouraging results. The most interesting solutions utilize image capture and analysis as vision-based body condition scoring systems, which somewhat mimics the traditional BCS assessment. Another imaging approach that has been used to measure body and carcass composition is ultrasound [115]. This technique has been widely used to monitor body condition in small ruminants [116,117], in swine [118], and in cattle [119]. For dairy cows, recent studies [120,121] showed the relevance of using ultrasound to assess the body reserves of cows with ultrasonic measurement to scan the body regions that are connected to the BCS evaluation, such as the ribs, pin, tail-head, and lumbar spine. However, despite the high accuracy for BCS prediction, the cows must be individually restrained to capture the ultrasound images, making this technique less suitable for analyzing large numbers of animals in multiple sessions over time. Therefore, this method is not appropriate for larger-scale farms with hundreds of animals. Consequently, the ultrasonic technique is reserved for punctual analyses or validation of other methods, such as those supported by cameras, where it is possible to obtain a BCS evaluation of animals in motion [122,123].

#### Vision-Based Body Condition Scoring Systems

Recently, a variety of vision-based solutions for BSC monitoring have been developed and tested, such as thermal imaging [122], 2D imaging [124], and 3D imaging technology [125,126]. Data analysis approaches have been applied to monitor the development of sensors, which increase the developed systems' capacity, with examples such as Fourier transformation [123] and machine learning [127]. However, despite the advances already made, there are still limitations to fully automated solutions. Nevertheless, with the development of cameras and software we are approaching objective and automatic BCS. The vision-based solutions remove the guesswork and imprecisions of conventional scoring, while the efficiency can be significantly improved. These reasons are certainly the basis for developing equipment that is well accepted by producers [128]. Table 3 summarizes research work assessing cow body condition score using 2D and 3D sensors.

Over the last decade, several researchers have focused their work on approaches with 2D cameras, but especially in recent years, attention has focused on 3D sensors, which have been widely applied to measure the energy reserves of dairy cattle [129]. 3D sensors have very different costs and typically use the time-of-flight (TOF) principle [130]. Several researchers, including Weber et al. [131], Spoliansky et al. [132], Alvarez et al. [133], Shigeta et al. [134], Hansen et al. [135], and Song et al. [136], used 3D sensors such as Microsoft Kinect or Asus Xtion2, which are related to gaming activities, and, therefore, aimed at reaching a vast market with a consequent decrease in sensor cost. Even so, 3D cameras are generally expensive, particularly those not incorporated in commercial solutions, which is understandable as the latter are subject to very challenging environments, which requires, in addition to the quality of the sensors, robust waterproof and dustproof equipment.

Making systems automatic is a necessary step to gain the interest of producers and thus turn the systems into a commercial business. To date, there are four automated BCS systems on the market [149]. All four systems use approaches based on image analysis captured from a 3D sensor placed on a higher plane of the rump and lumbar regions of the cows [149]. This is also the most common approach in non-commercial 3D and 2D solutions (Table 3). The commercial automatic BCS systems are DeLaval BCS (DeLaval International AB, Tumba, Sweden), BodyMat F (Ingenera SA, Cureglia, Switzerland), Biondi 4DRT-A (Biondi Engineering SA, Cadempino, Switzerland), and Protrack^®^ BCS (LIC Automation, Hamilton, New Zealand). The first commercially available system was the DeLaval BCS based on 3D image processing technologies; it was designed in 2015 by DeLaval Corporate [132]. The system operates while the cows move through a fixed point in the barn or on the DeLaval VMS™. The concept has made it feasible to incorporate BCS into herd management. The 3D camera is linked to a radio-frequency identification (RFID) system, which allows continuous monitoring of BCS and the use of this information in herd management systems [109]. A validation study has been conducted to examine the performance of the DeLaval BCS system [150]. This system was found helpful for automated monitoring of BCS variation. Moreover, the BCS camera system was reliable for cattle scored within the range of 3.00–3.75, where most cattle on the tested farm belonged, but did not score accurately with less than 3.00 and above 3.75. Furthermore, recently, an independent review of the BodyMatF BCS system has been published [149]. This work reached results similar to those obtained in the previous work, and allowed concluding that the automated and non-subjective nature of the BodyMatF system, combined with the ease of collecting regular scores, make this system likely to be of value in commercial and research contexts to evaluate Holstein-Friesian cow body condition. This technology can serve as a consistent source of BCS scores, which can be included in management processes and in the welfare assessment protocols. BCS has been included in the Welfare Quality protocols as an animal-based indicator linked to animal feed [151]. Similar to what is already in practice for other species (e.g., EyeNamic for Poultry and Swine [16]), PLF technologies have proven to be a step forward in the individual assessment of cows by continuous real-time monitoring of health and welfare [13,152].

## 3. The Potential of PLF for Assessing Welfare Animal-Based Indicators of Dairy Cattle

The assessment of the welfare of dairy cows, as well as other farm animal species involves audits that are time-consuming and expensive, as welfare is a complex multidimensional phenomenon [151]. On the other hand, with the advances that have been made in recent years in the use of sensor technologies, the main objective of PLF, which is the continuous real-time on-farm monitoring of individual animals to improve production/breeding, health and welfare, and environmental sustainability, is already being fulfilled in various aspects of dairy cattle production [152]. Regarding dairy cattle welfare assessment, as is the case with the Welfare Quality^®^ protocol, its application has meaningful constraints, as its application is very time-consuming [22] and lacks correspondence with trained users on the importance of several welfare measures [153]. In addition to reducing the evaluation time, several authors proposed some changes to the calculations, such as the one reported by Van Eerdenburg et al. [21] for drinking water. Moreover, the welfare calculations require more flexible methods, especially for the overall score [22,153]. That is why the possibility of applying PLF solutions to assess the animal-based indicators of lameness, mastitis, and body condition presented in this review will be very welcome. The advances discussed show that several PLF solutions have been developed and validated in recent years, and that is why there is the capacity to address the three animal-based indicators mentioned by commercial PLF technologies. Moreover, a recent review [12] pointed out that it will be necessary to modify some of the protocol criteria to take full advantage of the continuous measurement and individual monitoring of cows. This modification can rely on animal-based welfare measures, such as those analyzed in this paper and others, as suggested by Tuyttens et al. [22], who reviewed the Welfare Quality Protocol and found a more user-friendly, more time-efficient approach for assessing dairy cattle welfare, with the inclusion of only six animal-based indicators. There should also be room for other farm animal welfare frameworks, such as the five domains model [151]. The five domains model has gained interest among farm animal welfare researchers and has also been included in discussing the potential of applying the PLF to this model [154]. With the evolution of PLF solutions, the real-time monitoring of cow welfare supported by animal-based indicators is now undoubtedly feasible. Therefore, current scientific knowledge and technological development (e.g., Stygar et al. [13]) foresees important PLF developments in the near future, which will widen opportunities for assessing and improving the welfare of dairy cows.

## 4. Challenges for the Future

Precision livestock farming is recognized as fundamental for future dairy producers, allowing the continuous monitoring of the health and welfare of animals in production. In this review, the progress of exploiting technology for monitoring lameness, mastitis, and body condition in dairy cows is evident. For these problems, identified as animal-based indicators, accurate continuous monitoring systems, which avoid false alarms, are necessary for farmers to trust and adopt these technologies. Furthermore, to assess the welfare of dairy cows, a detailed early warning system is essential to prevent the development of more severe diseases and welfare problems. Finally, research into technology that ensures the welfare of dairy cows has provided several indicators that could be automatically measured and integrated into an assessment system.

## Figures and Tables

**Table 1 animals-11-02253-t001:** Summary of research work for assessing lameness of dairy cows by kinematic and kinetic approaches.

Approach	LS	n	Locomotion Test Layout	Results			Ref
				SE (%)	SP (%)	Accuracy (%)	
**Kinematic**							
Gaitwise	1–3	159	Alley 0.61 m wide and 4.88 m long	76–90	86–100		[42]
Gaitwise	1–3	40	Active surface of 0.61 m wide and 4.88 m long				[43]
Gaitwise	1–3	36	Active surface of 0.61 m wide and 4.88 m long	88	87		[38]
Gaitwise-14 configurations	1–3	45				55–61	[41]
3D Accelerometer	1–5	17 + 21		80–100	100	AUC = 0.87–1	[44]
**Kinetic**							
3D Accelerometer	1–5	12 + 36	Passageway (13 m long × 1.3 m wide)			>60	[45]
3D Accelerometer	1–5	17		100	75–83.3	AUC = 0.92–0.97	[44]
3D Accelerometer	1–5	21		83–91.7	66.7–83.3	AUC = 0.85–0.87	[44]
3D Accelerometer	1–5	348	Leg-mounted accelerometer				[46]
Ground force reaction	1–5	610	Stepmetrix system	35	85	–	[47]
Ground force reaction	1–5	83	Two parallel force plates	90	93	AUC = 0.98	[48]
Ground force reaction	1–5	105	Four-force plate-balanced system	50–100	91–100	–	[49]
Ground force reaction	1–5	95	Weight distribution of 4 limbs in milking robot			62–75	[50]
Ground force reaction	1–5	261	Two parallel force plates cow walks over	100	100	AUC = 0.70–0.99	[51]
Ground force reaction	1–5	346	Two parallel force plates cow walks over	52	89		[52]
Ground force reaction	1–5	43	Four sensor weight distribution of 4 limbs in milking robot				[53]
Ground force reaction	1–5	31	Two parallel force plates			0.84–0.63	[54]
Ground force reaction		6	Two parallel floor-plates plus SoftSeparatorTM				[55]
Ground force reaction	1–5	9	Two parallel 3D strain gauge force plates 0.46 m × 2.07 m	91–97			[56]
Ground force reaction		6	Two parallel floor-plates loading platform–126 × 122 × 18 cm				[57]
Load cells and platform	1–5	57	Four force plates cow stands on			AUC = 0.64–0.83	[58]
Load cells and platform	1–5	57	Four force plates cow stands on			AUC = 0.67	[59]
Load cells and platform	0–13	42	Platform with 4 independent sealed load cells	75–97	60–90	AUC = 0.84–0.87	[35]
Load cells and platform	1–5	16	Four-force plate-balanced system				[60]
Load cells and platform	1–5	73	Four force plates cow stands on	100	58	86–96	[61]
Motion sensor		10	Motion sensor attached hind left limb	74.2	91.6	91.1	[62]
Motion sensor		65	Dairy cow individual sensor			AUC = 0.71	[63]

LS, locomotion score; n, number of cows; SE, sensitivity = True Positive/(True Positive+False Negative) × 100; SP, Specificity = True Negative/(True Negative + False Positive) × 100; AUC, area under the curve; Ref, reference.

**Table 2 animals-11-02253-t002:** Summary of research works assessing the lameness of dairy cows using 2D and 3D sensors.

Image Equipment	LS	n	Setup	Results			Reference
				SE (%)	SP (%)	Accuracy (%)	
**2D**							
Canon Powershot A620	1–3	28	Alley (1.2 m wide and 6 m long)			>96	[67]
Guppy F-080C and Guppy F-036C	1–3	66	Alley (1.2 m wide and 6 m long)			>96	[67]
Guppy F-080C	1–3	75	Pressure mat (1 m wide and 6 m long)				[68]
Video Canon PAL MV690	1–5	60	Alley (1.6 m wide) electric fence posts				[69]
Cannon 60D	1–5	90	Alley (1.5 m wide and 7 m long)			76	[70]
Nikon D700	1–5	8	Alley (1.5 m wide and 7 m long)			91	[70]
Nikon D7000	1–5	273	Alley (1.1 m wide and 6 m long)	76–88	95–97	91–96	[71]
Web camera Hikvision	1–3	98	Alley (2 m wide and 7 m long)	90.25	94.74	90.18	[72]
Panasonic DC-GH5S	1–3	100	Alley (1.2 m wide and 4 m long)	93–96		96	[66]
Panasonic DC-GH5S	1–3	100	Alley (1.2 m wide and 4 m long)			93–96	[66]
**3D**							
Microsoft Kinect	1–5	186	3.20 m above ground level	55		90.9	[64]
Microsoft Kinect	1–5	273	3.15 m above ground level	82–88	91–95	90–96	[71]
Microsoft Kinect	1–5	242	3.45 m above ground level	68.5	87.6	79.8	[73]
Microsoft Kinect	1–5	242	3.45 m above ground level			70–72	[74]
Microsoft Kinect	1–5	270	3.45 m above ground level	74–72	60.2		[37]

LS, locomotion score; n, number of cows; SE, Sensitivity = True Positive/(True Positive + False Negative) × 100; SP, Specificity = True Negative/(True Negative + False Positive) × 100.

**Table 3 animals-11-02253-t003:** Summary of research work assessing cow body condition score using 2D and 3D sensors.

Sensor	n	Sensor Position	Accuracy	Accuracy within BCS Points Deviation (%)			Reference
				0	0.25	0.5	
**2D Sensors**							
Black-and-white	2571	60 to 70 cm above the cows’ backs			93	100	[137]
AXIS 213 PTZ	286	3 m above ground	Error = 0.31				[113]
InfraCAM SD Flir	186	3.1 m above ground. Exit milking parlor	R = 94				[122]
Nikon D7000 DSLR	151	Still camera-milking parlor	R^2^ = 77		50	100 ^#^	[124]
Sony, DCR-TRV460	46	3 m above ground	R^2^ = 90				[138]
Hikvision DS-2CD3T56DWD-I	8972	2.6 m the ground. Milking passage	R^2^ = 98.5				[106]
Hikvision DS-2CD3T56DWD-I	2231	Cows walk below the camera			65	95	[129]
**3D Sensors**							
Mesa 3D ToF	40	Hand-held setup			79	100	[139]
SR4K time-of-flight	540	Above electronic feeding dispenser	R^2^ = 89				[140]
ToF MESA SR4000	1329	Above DeLaval AWS 100	R = 84				[141]
Asus Xtion Pro	95	1.5–2m above the cow	R^2^ = 93.3				[142]
Asus Xtion Pro	82	2 m above ground	R = 96				[143]
Asus Xtion Pro	27	80 cm on cow’s surface	R^2^ = 74				[144]
PrimeSense™ Carmine	116	1.5 m from the cows’ backs			71	94	[145]
Microsoft Kinect v1	20	2.5 m above platform				91	[132]
Microsoft Kinect v2	1661	2.8 m above ground-milk parlor		40	78	94	[146]
Intel Realsense SR300	44	2.3 m above the platform	R^2^ = 72				[136]
Intel RealSense D435	480	3.2 m above ground			77	98	[147]
Microsoft Kinect v2	1661	2.8 m above ground-milk parlor			82	97	[133]
Microsoft Kinect v2	53	2.5 m above the ground	R^2^ = 63				[125]
Microsoft Kinect v2	38	3 m above the ground		56	76	94	[126]
3D ToF	52	3.4 m above ground-rotary parlor	MAPE = 3.9				[148]

n, number of cows; ToF, time of flight; BCS, body condition score; R, correlation coefficient; R^2^, coefficient of determination; MAPE, mean absolute percentage error; #, accuracy within 0.75 BCS points deviation.

## Data Availability

Not applicable.

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
