# Peer review of "Precision Technologies to Address Dairy Cattle Welfare: Focus on Lameness, Mastitis and Body Condition"

_animals, 2021, doi:10.3390/ani11082253_

Round 1

Reviewer 1 Report

The manuscript describes the importance of Technologies to Address Dairy Cattle Welfare: Focus on Lameness, Mastitis and Body Condition.
The review is interesting as the topic is very current and deals with a very important topic such as welfare in cattle breeding. In addition, it describes what are the most used technologies that can be used to detect lameness, mastitis and BCS. In addition, it also reports what may be the problems of each method used to detect the aforementioned parameters. However, animal welfare cannot only be measured by the lack of lameness, mastitis and BCS values. In the review it must be specified that these technologies together with other parameters that are detected on the animal and in the environment contribute to establishing the welfare state of the animal.
Therefore, by specifying better the function of PLF in regards to well-being, the work can be published in this Journal.However, animal welfare cannot only be measured by the lack of lameness, mastitis and BCS values. In the review it must be specified that these technologies together with other parameters that are detected on the animal and in the environment contribute to establishing the welfare state of the animal.
Therefore, by specifying better the function of PLF in regards to well-being, the work can be published in this Journal.

However, animal welfare cannot only be measured by the lack of lameness, mastitis and BCS values. In the review it must be specified that these technologies together with other parameters that are detected on the animal and in the environment contribute to establishing the welfare state of the animal.
Therefore, by specifying better the function of PLF in regards to welfare, the work can be published in this Journal.

Author Response

We appreciate the time and effort that you and the reviewers have dedicated to providing helpful feedback on our manuscript. The manuscript has been carefully revised and improved following the reviewer's comments. Likewise, we are grateful to the reviewers for their insightful comments on the paper. 

Reviewer 2 Report

animals-1298977

This review tries to summarize the available reports about automated detection methods for a number of elements of several welfare assessment protocols. Although they mention a few others in the introduction, the main assessment system that the authors refer to is the Welfare Quality system. Which is, in contrast to what the authors state in the manuscript, disputed in a number of ways and does not serve as a ‘gold standard’ in welfare assessment. Furthermore, several recent publications are missing.

Specific comments:

The English language used is, in general, acceptable, but needs to be improved. At many places there are repetitions.

Punctuation needs to be improved. It is lacking in many sentences.

There are too many references.

At several places in the text is mentioned: “an automated system may be able to identify animals with early onset of pathological or metabolic diseases and distress or discomfort, allowing early intervention by the farmer and improving animal health, production and welfare”. This doesn’t need to be repeated all these times.

Line 57-59 Sentence is not running well.

L 64 delete ‘of’.

L 71-73. The Welfare quality system has been seriously disputed and is not well accepted by everyone. See for example:

  1. Tuyttens, F.A.M.; De Graaf, S.; Andreassen, S.N.; De boyer Des Roches, A.; Van Eerdenburg, F.; Haskell, M.J.; Kirchner, M.; Mounier, L.; Kjosevski, M.; Bijttebier, J., et al. Using expert elicitation to simplify the Welfare Quality® protocol for monitoring the most adverse dairy cattle welfare impairments. Frontiers in Veterinary Science 2021, 10.3389/fvets.2021.634470, doi:10.3389/fvets.2021.634470. (This article is included as reference number 161)
  2. Van Eerdenburg, F.J.C.M.; Di Giacinto, A.M.; Hulsen, J.; Snel, B.; Stegeman, J.A. A new, practical animal welfare assessment for dairy farmers. . Animals 2021, 11, 881, https://doi.org/10.3390/ani11030881.
  3. De Vries, M.; Engel, B.; Den Uijl, I.; Van Schaik, G.; Dijkstra, T.; De Boer, I.M.; Bokkers, E.A. Assessment time of the Welfare Quality® protocol for dairy cattle. Animal Welfare 2013, 22, 85-93.
  4. De Vries, M.; Bokkers, E.A.; van Schaik, G.; Botreau, R.; Engel, B.; Dijkstra, T.; de Boer, I. Evaluating results of the Welfare Quality multi-criteria evaluation model for classification of dairy cattle welfare at the herd level. J Dairy Sci 2013, 96, 6264-6273.
  5. Bokkers, E.A.; De Vries, M.; Antonissen, I.C.M.A.; De Boer, I.M. Inter- and intra-observer reliability of experienced and inexperienced observers for the Qualitative Behaviour Assessment in dairy cattle. Animal Welfare 2012, 21, 307-318.

L 77 It is not the intention of a welfare assessment system to ‘predict’ behaviour, but to ‘record’ it.

L 81 ‘provide’.

L91 ‘impaires’.

L 146: “this action is thought to be painful. “ The reason that a cow walks irregular is because it is painful to place her feet on the ground. So you may delete ‘thought to be’.

L 154 ‘reports’.

L 186: Most pedometers are accelerometers these days.

L 226-228: This message has been stated several times now.

L 232: this is the first time ‘thermal’ data are mentioned. So why the ‘other’?

L 236: This paragraph is not very clear and has a number of repetitions.

L 244-250: Please rewrite this part.

L 305: This has been stated in line 301 already.

L306-307 & 415-418: This is a general statement that applies to the entire review. And this has been mentioned several times before.

L326: ‘approach’, and delete ‘is’.

L345 & 359: ‘systems’.

L 428: I did not read in this publication that the Welfare Quality protocol is widely accepted. It suggests to modify the protocol, because it is not.

L 429: The Welfare Quality is protocol is not considered a success. In ref 161 is stated: “The uptake of the dairy cattle protocol has been below expectation, however, and it has been criticized for the variable quality of the welfare measures and for a limited number of measures having a disproportionally large effect on the integrated welfare categorization.”.

L 436: ‘points’

L 447: ‘development’.

  1. Conclusions: These are not conclusions of the reviewed literature. These are more the challenges for the future.

Author Response

(The authors gave the same response as above.)

Round 2

Reviewer 2 Report

Comments from Reviewer 2

The authors made some adjustments to the manuscript. However, one of my major concerns with this manuscript is the interpretation of the authors of the Welfare Quality protocol. As I mentioned in my previous report, this protocol is not well accepted as the authors stated. They made some changes accordingly, but still fail to point at all the shortcomings. One of those shortcomings is, indeed, the time needed to perform one assessment. And this can be overcome by the use of sensors. However, the other major concern with this protocol is the way the measures are evaluated and the final outcome is calculated. De Graaf et al. (ref 151) dedicated their entire publication to this problem. The authors, however, interpret this completely different and cite this paper as ‘supporting’ the calculations of Welfare Quality.

Below I give a reply to the answers of the authors if needed. When there is no reply, I agree.

Previous report, answers of the authors and my answers:

This review tries to summarize the available reports about automated detection methods for a number of elements of several welfare assessment protocols. Although they mention a few others in the introduction, the main assessment system that the authors refer to is the Welfare Quality system. Which is, in contrast to what the authors state in the manuscript, disputed in a number of ways and does not serve as a ‘gold standard’ in welfare assessment. Furthermore, several recent publications are missing.

Author’s response: The authors are grateful for the comment, which we hope is now addressed with the changes introduced throughout the manuscript to answer specific questions or clarify the text. Please see text lines 70 to 90; 238 to 247; 296 to 300; 417-422; 429 to 433.

Reviewer’s answer: The lines indicated do not point to alterations in the text. Only at lines 70 to 90 I see new text but for example at lines 238-247: Since behavior assessment can be a long-time task, the use of technology is crucial [16]. Change in an animal's behaviour is one of the most important criteria in assessing animal welfare and health. For example, pain associated with claw or limb disorders cause alterations in gait characteristics and a decreased daily activity level [75]. Additional placement sensor type in the same body location (e.g., rumination audio sensor, magnetometer) or an additional accelerometer in an alternative body location (e.g., leg-mounted) would likely be needed to accurately classify the three main behaviours of interest in dairy cows (lying, standing, and feeding) [29,82]. Analysis of the classified behaviour cohort highlights differences in feeding activity, with feeding duration being significantly lower for lame cows than non-lame cows. This is the same text as before and deals not with critical remarks for the Welfare Quality protocol. At lines 296 to 300; 417-422; 429 to 433 this is the same.

In the new text  at lines 70-90 it is mentioned that: “because the associated assessment protocol is time-consuming, limiting its routine application in dairy farms” as the only criticism. The evaluation and values of certain measures in the WQ protocol are also points of concern. This should be mentioned here as well. This is important because one might be able to replace the time consuming observations with sensor data, but the evaluation and calculations are problematic too. See for example:

Heath, C.A.; Browne, W.J.; Mullan, S.; Main, D.C. Navigating the iceberg: reducing the number of parameters within the Welfare Quality(®) assessment protocol for dairy cows. Animal 2014, 8, 1978-1986, doi:10.1017/S1751731114002018.

Toma, L.; Haskell, M.J.; Barnes, A.P.; Stott, A.W. Relationship between animal welfare, production and environmental performance of dairy farms. In Proceedings of 7th International Conference on the Assessment of Animal Welfare at Farm and Group Level, Ede, the Netherlands, 2017; p. 39.

And De Graaf et al. (now ref 151)

Specific comments: The English language used is, in general, acceptable, but needs to be improved. At many places there are repetitions.

Author’s response: The entire manuscript has been revised, and the text that was repeated has been changed or deleted.

Reviewer’s answer: the repetitions still exist.

Punctuation needs to be improved. It is lacking in many sentences.

Author’s response: The entire text was revised with care for punctuation, with changes throughout the manuscript in the revised version. Also, all spelling and grammatical errors pointed out by the reviewer have been corrected.

Reviewer’s answer: this has been improved indeed.

There are too many references.

Author’s response: The authors appreciate the comment. Some references have been removed throughout the manuscript. The respective numbering in the text and the references section has been changed accordingly. Now the revised version has 152 references. References deleted

29Adams, A.E.; Lombard, J.E.; Fossler, C.P.; Román-Muñiz, I.N.; Kopral, C.A. Associations between housing and management practices and the prevalence of lameness, hock lesions, and thin cows on US dairy operations. J. Dairy Sci. 2017, 100, 2119-2136. 10.3168/jds.2016-11517

31Groenevelt, M.; Main, D.C.J.; Tisdall, D.; Knowles, T.G.; Bell, N.J. Measuring the response to therapeutic foot trimming in dairy cows with fortnightly lameness scoring. Vet. J. 2014, 201, 283-288. 10.1016/j.tvjl.2014.05.017

81 Meagher, R.K.; Beaver, A.; Weary, D.M.; von Keyserlingk, M.A.G. Invited review: A systematic review of the effects of prolonged cow–calf contact on behavior, welfare, and productivity. J. Dairy Sci. 2019, 102, 5765-5783. 10.3168/jds.2018-16021

85 Van Soest, F.J.S.; Santman-Berends, I.; Lam, T.; Hogeveen H. Failure and preventive costs of mastitis on Dutch dairy farms. J. Dairy Sci. 2016, 99, 8365-8374. 10.3168/jds.2015-10561

89 González Pereyra, V.; Pol, M.; Pastorino, F.; Herrero, A. Quantification of antimicrobial usage in dairy cows and preweaned calves in Argentina. Prev. Vet. Med. 2015, 122, 273-279. 10.1016/j.prevetmed.2015.10.019

92 Lehmann, M.; Wall, S.K.; Wellnitz, O.; Bruckmaier, R.M. Changes in milk L lactate, lactate dehydrogenase, serum albumin, and IgG during milk ejection and their association with somatic cell count. J. Dairy Res. 2015, 82, 129-134. 10.1017/S002202991400065X

106Metzner, M.; Sauter-Louis, C.; Seemueller, A.; Petzl, W.; Zerbe, H. Infrared thermography of the udder after experimentally induced Escherichia coli mastitis in cows. Vet. J. 2015, 204, 360-362. 10.1016/j.tvjl.2015.04.013

107Metzner, M.; Sauter-Louis, C.; Seemueller, A.; Petzl, W.; Klee, W. Infrared thermography of the udder surface of dairy cattle: characteristics, methods, and correlation with rectal temperature. Vet. J. 2014, 199, 57-62. 10.1016/j.tvjl.2013.10.030

110Akbar, H.; Grala, T.M.; Vailati, R.M.; Cardoso, F.C.; Verkerk, G.; McGowan, J.; Macdonald, K.; Webster, J.; Schutz, K.; Meier, S.; Matthews, L.; Roche, J.R.; Loor, J.J. Body condition score at calving affects systemic and hepatic transcriptome indicators of inflammation and nutrient metabolism in grazing dairy cows. J. Dairy Sci. 2015, 98, 1019-1032. 10.3168/jds.2014-8584

121Teixeira, A.; Silva, S.; Rodrigues, S. Advances and sheep and goat meat products research. In Advances in Food and Nutrition Research; Toldra, F., Ed.; Academic Press: Cambridge, MA, USA, 2019; Volume 87, pp. 305-370.

126Hussein, H.A.; Westphal, A.; Staufenbiel, R. Relationship between body condition score and ultrasound measurement of backfat thickness in multiparous Holstein dairy cows at different production phases. Aust. Vet. J. 2013, 91, 185-189. 10.1111/avj.12033

127Termatzidou, S.A.; Siachos, N.; Valergakis, G.E.; Georgakopoulos, A.; Patsikas, M.N.; Arsenos, G. Association of body condition score with ultrasound backfat and longissimus dorsi muscle depth in different breeds of dairy sheep. Lives. Sci. 2020, 236, 104019. 10.1016/j.livsci.2020.104019

At several places in the text is mentioned: “an automated system may be able to identify animals with early onset of pathological or metabolic diseases and distress or discomfort, allowing early intervention by the farmer and improving animal health, production and welfare”. This doesn’t need to be repeated all these times.

Author’s response: This comment was greatly appreciated. The manuscript has been revised, and changes have been made throughout the manuscript to eliminate or alter the text not to repeat it. As a result, we think that the text now has a more fluent reading.

Line 57-59 Sentence is not running well.

Author’s response: The text was changed. Please see Lines 57 to 59.

L 64 delete ‘of’.

Authors response: The word “of” was deleted

L 71-73. The Welfare quality system has been seriously disputed and is not well accepted by everyone. See for example:

  1. Tuyttens, F.A.M.; De Graaf, S.; Andreassen, S.N.; De boyer Des Roches, A.; Van Eerdenburg, F.; Haskell, M.J.; Kirchner, M.; Mounier, L.; Kjosevski, M.; Bijttebier, J., et al. Using expert elicitation to simplify the Welfare Quality® protocol for monitoring the most adverse dairy cattle welfare impairments. Frontiers in Veterinary Science 2021, 10.3389/fvets.2021.634470, doi:10.3389/fvets.2021.634470. (This article is included as reference number 161)
  2. Van Eerdenburg, F.J.C.M.; Di Giacinto, A.M.; Hulsen, J.; Snel, B.; Stegeman, J.A. A new, practical animal welfare assessment for dairy farmers. . Animals 2021, 11, 881, https://doi.org/10.3390/ani11030881.
  3. De Vries, M.; Engel, B.; Den Uijl, I.; Van Schaik, G.; Dijkstra, T.; De Boer, I.M.; Bokkers, E.A. Assessment time of the Welfare Quality® protocol for dairy cattle. Animal Welfare 2013, 22, 85-93. 4. De Vries, M.; Bokkers, E.A.; van Schaik, G.; Botreau, R.; Engel, B.; Dijkstra, T.; de Boer, I. Evaluating results of the Welfare Quality multi-criteria evaluation model for classification of dairy cattle welfare at the herd level. J Dairy Sci 2013, 96, 6264-6273.
  4. Bokkers, E.A.; De Vries, M.; Antonissen, I.C.M.A.; De Boer, I.M. Inter- and intra-observer reliability of experienced and inexperienced observers for the Qualitative Behaviour Assessment in dairy cattle. Animal Welfare 2012, 21, 307-318.

Author’s response: The authors appreciate the comment. The authors agree that the Welfare Quality protocol has been challenged and hence the emergence of new approaches with other indicators or a reduction in the number of indicators, electing those that most contribute to welfare evaluation; they are ways to assess cow welfare in farm context more effectively. Please see lines 70 to 75.

L 77 It is not the intention of a welfare assessment system to ‘predict’ behaviour, but to ‘record’ it. Authors response: the text is changed

L 81 ‘provide’.

Authors response: the text is changed

L91 ‘impaires’.

Authors response: the word is changed

L 146: “this action is thought to be painful. “ The reason that a cow walks irregular is because it is painful to place her feet on the ground. So you may delete ‘thought to be’.

Authors’ response: We agree with the reviewer’s comment. “thought to be” was deleted.

L 154 ‘reports’.

Authors response: the word is changed

L 186: Most pedometers are accelerometers these days.

Authors response: The text is changed.

L 226-228: This message has been stated several times now. Author’s response: The authors agree. As mentioned in other authors' responses, this comment was greatly appreciated, and the text changed throughout the manuscript for this issue.

L 232: this is the first time ‘thermal’ data are mentioned. So why the ‘other’?

Author’s response: The authors agree. Thermal word is deleted

L 236: This paragraph is not very clear and has a number of repetitions.

Author’s response: The text has been revised.

L 244-250: Please rewrite this part. Author’s response: The text has been rewritten. Please see lines 238 to 247.

L 305: This has been stated in line 301 already.

Author’s response: The text has been rewritten.

L306-307 & 415-418: This is a general statement that applies to the entire review. And this has been mentioned several times before.

Author’s response: Authors agree. Text revised or deleted.

L326: ‘approach’, and delete ‘is’.

Author’s response: The text was changed

L345 & 359: ‘systems’.

Author’s response: The word was changed

L 428: I did not read in this publication that the Welfare Quality protocol is widely accepted. It suggests to modify the protocol, because it is not.

Author’s response: The authors acknowledge that the reference to Tuyttens et al. 2021 is not suitable for this sentence. The text has been revised and a more suitable reference introduced.

Reviewer’s answer: the new reference introduced here is from the same group and confirms the previous statement about the fact that WQ is not accepted by many experts. In ref 151 it is stated: “The level of correspondence between expert scoring and WQ scoring for 6 of the 12 criteria and for the overall welfare score was low. The WQ scores of the protocol for dairy cattle thus lacked correspondence with trained users on the importance of several welfare measures.”

Please see Line L 429: The Welfare Quality is protocol is not considered a success. In ref 161 is stated: “The uptake of the dairy cattle protocol has been below expectation, however, and it has been criticized for the variable quality of the welfare measures and for a limited number of measures having a disproportionally large effect on the integrated welfare categorization.”.

Author’s response: Authors agree and the text was changed.

L 436: ‘points’ Author’s response: The word was changed

L 447: ‘development’.

Author’s response: The word was changed.

Conclusions: These are not conclusions of the reviewed literature. These are more the challenges for the future.

Author’s response: The authors appreciate the reviewer's comment and the text has been changed.

Author Response

The authors acknowledge the reviewer comments, and changes were made throughout the text to address their comments.

Round 3

Reviewer 2 Report

Just a few remarks left:

Line 72-74: The authors still do not get my point about the problems with the Welfare Quality protocol. It is not only the time consuming observations that are a problem, but also the way of assessing and calculating the scores. In the paper of Van Eerdenburg et al., the procedures are shortened, but also some of the calculations changed (specifically drinking water). In the paper of Tuyttens et al. and De Graaf et al. the calculations are changed, especially the overall score. This is important because if you could reduce the time needed for the execution of the protocol, the calculations still remain a problem. So by mentioning only the problems with the execution time of the WQ protocol, you miss an important part. I do realize that the authors have no suggestions for changes in this matter, but they should mention this at least here and in the discussion.

L 73: reducing

L74: Insert ‘observations’ after ‘time-consuming’.

Line 92: According to Huxley (Huxley, J.N. Impact of lameness and claw lesions in cows on health and production. Livest Sci 2013, 156, 64-70, http://dx.doi.org/10.1016/j.livsci.2013.06.012.) the reduction in milk yield is mainly due to less eating. My suggestion is to delete the remark about the reason for reduction in milk yield.

Author Response

According to the comments of reviewer, the improvements have been incorporated in the article.
